# Short- and Long-Term Effects of Split-Suckling in Pigs According to Birth Weight

**DOI:** 10.3390/ani13223521

**Published:** 2023-11-14

**Authors:** María Romero, Luis Calvo, José Ignacio Morales, Antonio Magro, Ana Isabel Rodríguez, José Segura, Rosa Escudero, Clemente López-Bote, Álvaro Olivares

**Affiliations:** 1Departamento de Producción Animal, Facultad de Veterinaria, Universidad Complutense, Avda. Puerta de Hierro s/n, 28040 Madrid, Spain; maria.romero@copiso.com (M.R.); amagro01@ucm.es (A.M.); josesegu@ucm.es (J.S.); rmescude@ucm.es (R.E.); clemente@ucm.es (C.L.-B.); 2Copiso, Avda. de Valladolid, 105, 42005 Castilla y León, Spain; ji.morales@copiso.com; 3Incarlopsa, Ctra. N-400, Km. 95,4, 16400 Castilla La Mancha, Spain; luiscalvo@incarlopsa.es (L.C.); airodriguez@incarlopsa.es (A.I.R.)

**Keywords:** piglet, split-suckling, colostrum, birth weight, meat quality

## Abstract

**Simple Summary:**

Increasing prolificity in swine reduces the opportunities of pigs to obtain enough amount of colostrum, which may lead to negative short- and long-term consequences. This is especially important for piglets with a low birth weight. Removing the heaviest piglets from the litter for a short period during suckling (split-suckling) may allow the smallest and less viable piglets to have access to the udder during the critical initial hours after farrowing. This experiment has shown that mortality is not altered by split-suckling. Moreover, estimation of colostrum intake suggests that the heaviest piglets subjected to split-suckling ingest a lower amount and show a lower weight gain during lactation than their litter mates. We have not observed differences in final weight, but split-suckling increases carcass leanness, especially in piglets of low birth weight. Differences in the fatty acid composition of intramuscular lipids because of split-suckling were also observed. Long-term response to split-suckling, particularly in low-birth-weight piglets, suggests an alteration in adiposity and metabolic regulation which may be related to higher colostrum intake.

**Abstract:**

Forty-eight litters were used, with a total number of 645 piglets involved in the study. The split-suckling technique was applied to half of the litters at the end of farrowing by removing the heaviest piglets over three periods of 1 h. The piglets were individually weighed at 0, 1 d, and at weaning. Piglet losses were recorded daily. Traceability was maintained until the carcass splitting and meat analysis took place. Carcasses were eviscerated and weighed individually. Total mortality at weaning was affected by body weight, where the low-body-weight piglets showed a mortality rate almost four times higher than that of the normal-weight piglets. Mortality was highest in the first days of life, especially in the low-body-weight piglets. At weaning, split-suckling treatment caused a slight increase in mortality compared to the control group piglets (25% vs. 17.1%). Split-suckling had a positive effect on weight gain during the first 24 h of life (*p* = 0.014), and there was an interaction between treatment and parity (*p* = 0.007), with split-suckling being more effective in the primiparous sows compared to the multiparous sows. The piglets from litters receiving the split-suckling treatment had a lower average daily gain during the lactation period (*p* < 0.001) than the piglets from the control group. Weight gain during the first 24 h of life of the piglets subjected to split-suckling was higher than those of the control group. A lower IgG and α-tocopherol in plasma in the heavier piglets subjected to split-suckling treatment was observed in comparison to their respective control. The piglets from litters receiving the treatment showed a lower average daily gain during the lactation period (*p* < 0.001) than the piglets from the control group. No difference in slaughter weight was observed according to treatment. The pigs which received split-suckling treatment showed lower subcutaneous fat thickness (*p* < 0.0013) and higher lean meat yield (*p* < 0.0027), this effect being more marked in pigs from primiparous sows. Intramuscular fat concentration was higher in the *Longissimus Dorsi* muscle of the low-body-weight piglets. In the pigs that received split-suckling treatment, a higher concentration of C18:3n-3 (*p* = 0.036) and a tendency towards a higher concentration of C18:2n-6 (*p* = 0.107) and unsaturation index (*p* = 0.113) was observed in intramuscular fatty acids at slaughter, together with a lower concentration of C16:0 (*p* = 0.053) and SFA (*p* = 0.064). In conclusion, long-term response to split-suckling, particularly in low-birth-weight piglets, suggests an alteration in adiposity and metabolic regulation in these piglets that receive high levels of colostrum.

## 1. Introduction

Genetic improvements in prolificacy have resulted in a marked increase in the number of piglets born per litter [1,2], which indirectly causes an increase in the proportion of piglets with low birth weight (BW). Since colostrum production is not influenced by the number of piglets born, the larger the litter, the greater the competition for colostrum intake [3]. Thus, inadequate colostrum intake is considered one of the causes of higher preweaning mortality in hyperprolific sows [4]. Devillers et al. [5] estimated that colostrum intake below 250–290 g (equivalent to 20% of live weight) negatively affects piglet growth before weaning and observed a preweaning mortality of 43% in piglets with colostrum intake below 200 g, and only 7% when intake increased. Apart from providing essential elements like water, electrolytes, protein, energy and immunoglobulins, colostrum contains a series of active compounds such as relaxin, hormones, and cytokines. These active compounds can have a long-term effect on growth and body composition, which can affect productivity in later stages of the piglet live [6]. They might also have effects on carcass and meat characteristics, although there is limited scientific information available regarding these later aspects.

It is not easy to estimate colostrum intake directly, since it requires double weighing of the litter before and after each lactation episode, which is an inadvisable management practice at such a delicate time as postpartum. Indirect procedures based on the individual weight gain of each piglet during the first 24 h of life can also be used [5]. This procedure is the most commonly used in practice, although it may result in high variability. This is because piglets that consume colostrum tend to be the most vigorous at birth, and thus, the effects observed during their productive life may not be exclusively attributed to colostrum intake. Immunoglobulin concentration in blood, at 7 days or at weaning, is also used as an indirect measure of colostrum intake, although the results are inconclusive [7].

To manage low colostrum intake in piglets with reduced chances of survival, different management techniques are utilized. One of them is split-suckling (SS) [8], which consists of limiting access to the udder for some litter mates to allow low-body-weight piglets to acquire colostrum. There is no consensus on how to apply split-suckling, in terms of duration, which piglets should be isolated, number of groups to make, group sizes, and when to start [9]. Several possible protocols may include separating only some specific piglets (heavier piglets, or those that have been verified to have ingested colostrum) and establishing two groups in the litter and allowing access to the udder alternately [10,11]. The split-suckling technique is not widespread, but in some countries it is used by an increasing number of farmers, especially in the case of hyperprolific sows, reaching figures close to 40% of farms [8], although there are very few scientific articles presenting contrasted results that support the validity of this procedure when studying productive data of the pig production sector [9,10,12]. Morton et al. [7] and Huser et al. [13] have observed that the separation of only the heaviest pigs allows a higher concentration of immunoglobulins and an improvement in survival in small-sized piglets. Therefore, it could be inferred that intervening during the neonatal stage of piglets, to ensure an optimal colostrum intake, might result in outcomes that go beyond reducing mortality rates, which could support the adoption of this management practice in the swine industry.

This research was designed to study how birth weight and split-suckling treatment administered to newborn piglets affect preweaning mortality and, in the long term, on growth and carcass characteristics, and the meat quality parameters.

## 2. Materials and Methods

### 2.1. Animals

This research was carried out at a COPISO farm (Cubo de la Solana, located at the province of Soria, north of Spain) between January and July 2020. Housing was provided under directives of the EU (European Directive 2008/120/EC). Also, any invasive procedures on the animals were developed in compliance with European Directive 2010/63/EC Article 1 5. (f). The study was evaluated and accepted by the Animal Welfare Committee of the *Comunidad of Castilla y León* (Valladolid, Ref 10052022). Overall, 48 crossbred sows (Landrace × Large White, PIC 32) were considered. A total of 48 litters of piglets were used, 13 from primiparous gilts (PP) and 35 from multiparous sows (MP), with 645 piglets involved in the study. Heterospermic inseminations were performed using recently collected semen from Danbred Sires. Semen was stored in a refrigerated environment and utilized within the initial 48 h after collection.

### 2.2. Farrowing, Split-Suckling Treatment, and Lactation

During gestation, both sows and gilts were kept in groups of 60 in gestation pens. At one week prior to the expected farrowing date, the pregnant sows were transferred to pens with farrowing crates equipped with fully slatted floors. Two rooms, located in the same building, were needed for each farrowing system. The farrowing box was 2.6 m long and 2.0 m wide, and the dimensions of the farrowing crate were 1.9 m × 0.80 m. Sows whose farrowing duration was longer than 3 h were eliminated from the study.

A peripartum diet (16.5% CP, 2.15 Mcal EN/kg, and 3.95 g SID Lys/Mcal EN) was fed restrictedly twice a day (7 a.m. and 3 p.m.) at 3 kg/sow/d from gestation day 112 to 4 d after farrowing (based on the sow farrowing date). After farrowing, diets were provided at 3.5, 4.0, 4.5, and 5.0 kg/sow/d. Starting at day 5 of lactation to weaning, a commercial lactation diet (16.6% CP, 2.3 Mcal EN/kg, and 3.69 g SID Lys/Mcal EN) was provided using a Spotmix system (Schauer-Agrotronic, Prambachkirchen, Austria) by increasing 0.5 kg/sow/d until maximum voluntary intake. Sows had *ad libitum* access to drinking water. Piglets were given access to nipple drinkers and were not offered creep feeding. To minimize variation sources, the same daily work routine was kept.

Upon birth, each piglet was individually weighed and identified with electronic ear tags (MPIG-DATA, Madrid, Spain). At farrowing, piglets were distributed according to their birth weight (BW) and considering the number of functional teats, and therefore, fostering. Piglets weighing over 950 g were considered normal BW, and the others were classified as low BW. At 7 d postnatal, blood samples were collected in EDTA tubes.

The split-suckling (SS) technique was applied to half of the litters (24 of the 48) on the first day of birth. The protocol followed was divided into several phases:At the end of farrowing (when the sow expelled the placenta), normal-BW piglets were separated for 1 h to allow access for the low-BW piglets. Piglets removed from the litter were calculated according to the number of functional teats.After the first hour, the piglets that had been removed in step 1 were reintroduced among the piglets remaining with the sow, and another set of heavy piglets different from the first selection was removed for 1 h.After the second hour, the procedure was repeated by removing those piglets that were selected the first time (step 1).Finally, after the third hour, the last piglets removed were reintroduced so that the litter was complete for the rest of the lactation.

Piglets were individually weighed at 0, 1 d of life, and at weaning. Piglet deaths were recorded daily. Fostering was carried out at 24 h of life to allow equal number of piglets to functional teats. Piglets with weights close to the mean value of the litter were removed when necessary. Weaning was performed at an average of 25.8 ± 1.9 d, with no variations based on parity.

### 2.3. Weaning, Growing, and Fattening

At weaning, piglets were kept in groups of 24 animals. Flat-deck pens (2.60 m × 2.35 m) were used with two water drinkers (bowl) and a unique feeder (1.5 m length). During the first two weeks, temperature was set at 30 ± 2 °C, and then it was progressively decreased by 2 °C weekly until the temperature of 24 °C was reached. The pigs had unrestricted access to both feed and water throughout the duration of the experiment. Diet and handling were identical for all animals in both the treatment and control groups. The initial 2 weeks after weaning, a diet was provided which contained 2.52 Mcal EN/kg, 18.5% CP, and 5.59 g SID Lys/Mcal EN. The following three weeks, a starter diet was administered, which contained 2.45 Mcal EN/kg, 18% CP, and 5.38 g SID Lys/Mcal EN). In all cases, water and diets were provided for *ad libitum* intake.

All replicate pens (*n* = 50) had a similar average BW and equal proportion of males and females. The assignment of pigs to replicates did not take into account the origin of the litter. Upon arrival at the growing–finishing farm, the pigs were accommodated in a naturally ventilated finishing barn. They were placed in pens measuring 3.1 m^2^ × 3.1 m, featuring 80% slatted concrete floors, with 13 pigs per pen. Experienced staff examined the pigs daily, and the veterinarian in charge of the farm revised the facilities and pigs weekly. Feed was provided in pelleted form. Main ingredients were cereals and soybean meal and diets were formulated to meet the nutrient requirements of pigs [14]. Five diets were formulated, which contained, respectively, 2.4, 2.4, 2.55, 2.55, and 2.55 kcal EN/kg; 17.7, 16.9, 14.6, 12, 3, and 11.2% CP; 4.5, 4.2, 3.36, 2.7, and 2.3 g SID Lys/Mcal EN, for pigs in the ranges of 20–30, 30–45, 45–60, 60–90, and 90–120 kg. All pigs (n = 645) were transported to the abattoir the same day (with 170–175 d of age).

### 2.4. Carcass and Meat Quality

The pigs were transported 300 km to a commercial abattoir (Incarlopsa, Tarancón, Cuenca, Spain). Upon arrival, they were given a 5-h lairage period to rest, and they were fasted for a total of 12 h before processing. Pigs were stunned in a 95% CO_2_ atmosphere, then exsanguinated and scalded at 65 °C according to standard commercial procedures. Total lean meat yield and ham lean and fat percentages were measured using the AutoFom III classification system (Carometec, Barcelona, Spain).

After evisceration, carcass weight was recorded. Afterwards, the head was removed at the atlanto-occipital junction and the carcasses were cooled at approximately 2 °C for 2 h (air speed of 1 m/s and 90% relative humidity). Subcutaneous fat at the level of the 3–4th last ribs 6–8 cm from the midline (P2) and on external ham, was measured in each carcass. Muscle ultimate pH (pHu) was measured by means of a K21 pH meter (NWK Thien, Landsberg, Germany).

Carcass cut-out was performed according to the simplified European Union reference method [15]. Hams and loins were removed and kept in the chilled room at 4 °C for 24 h and then were weighed.

Samples of approximately 200 g of *Semimembranosus* (SM), *Biceps femoris* (BF), and *Longissimus dorsi* (LD) muscle (at the level of the last rib) were collected from 96 pigs, 6 per experimental group (2 split-suckling treatment × 2 parity × 2 sex × 2 BW), and placed in individual plastic bags. Intramuscular fat content (FOSS 6500 Spectrophotometer; Foss Analytical, Barcelona, Spain) was determined.

Color was measured by a Chroma Meter (CM-2002, Minolta device, Osaka, Japan) which was previously calibrated with a white tile (CIE, 1976). A D65 illuminant, observer 2° in SCI mode, and 1 cm aperture was used. The mean value of five measurements was used to calculate lightness (*L**), redness (*a**), and yellowness (*b**).

Meat samples used to quantify intramuscular fat and fatty acid composition were kept vacuum packaged and frozen at −20 °C until analyses. Intramuscular fat was extracted and methylated according to Segura and Lopez-Bote [16] in freeze-dried samples. The final biphasic system was separated by centrifugation (8 min at 10,000 rpm), the solvent was evaporated under a nitrogen stream, and the lipids remaining were quantified gravimetrically.

Lipids were esterified by heating at 80 °C for 1 h in 3 mL of 88:10:2 methanol-toluene-H_2_SO_4_. Fatty acids were identified and measured using gas chromatography (6890 Hewlett Packard, Avondale, PA, USA). The process involved an automatic injector maintained at 250 °C and a flame ionization detector. The fatty acid (FA) methyl esters were separated in a capillary column (HP-INNOWax, 30 m × 0.32 mm × 0.25 μm) (Agilent Technologies GmbH, Wald-bronn, Germany) using a temperature gradient of 170 °C to 245 °C. A split ratio of 1:50 was used. Identification of each individual FA was carried out by using standards (Sigma-Aldrich, Madrid, Spain). FA concentration was expressed as percentage of the total FAs. The Δ9 desaturase index was calculated as = (C16:1n-7 + C18:1n-9) × 100/(C16:0 + C16:1n-7 + C18:0 + C18:1n-9). The average number of double bonds per FA residue was used to calculate unsaturation index (UI).

Plasma α-tocopherol concentration was analyzed by HPLC as described by Rey et al. [17]. Plasma immunoglobulin G (IgG) concentration was determined by absorbance change when mixed with goat anti-IgG antibodies quantified by comparison from a calibrator of known IgG concentration (SPINREACT S.A., Girona, Spain).

### 2.5. Statistical Analysis

Statistical analysis was carried out using the software package SAS 9.4 (SAS Institute Inc., 229 Cary, NC, USA). Data were analyzed by Shapiro–Wilk and Levene’s test to determine normality and variance homogeneity. Mortality data did not follow normal distribution and was analyzed by the non-parametric Chi-square procedure. To study differences in production and carcass characteristics, a completely randomized design using the general linear model (GLM) was used to analyze the data procedure. The sow was the random effect in the statistical model. Fixed effects were split-suckling treatment (T), parity (P), and sex (S). For carcass and splitting yield, carcass weight was included as the covariable in the model. Birth weight class was not included as a fixed effect because the data did not follow normal distribution. In this case, the effect of BW was analyzed by using linear regression and slope comparison, which were compared by Student’s t-test. Meat quality characteristics followed a normal distribution in all cases and, therefore, T, P, S, and BW classes were included as fixed effects. Data are presented in tables as the mean (LS means for carcass characteristics) of each group, the pooled standard deviation (SD), and the *p*-value of the main effects and interactions. When *p* < 0.05, differences between treatment means were considered statistically significant.

## 3. Results

### 3.1. Mortality

The effect of birth weight, split-suckling, parity, and sex on mortality during the first 6 days of life and along the whole lactation period is shown in Table 1. Mortality was considered at the first days of life, particularly in the low-BW piglets. Regarding total weaning mortality, the low-BW piglets showed nearly a four-fold higher value than the piglets of normal weights (55.4% vs. 15.7%) (*p* = 0.0001). At weaning, SS treatment caused a slight increase in mortality compared to the control group piglets (25% vs. 17.1%). No effect of parity or sex was observed for this parameter.

### 3.2. Colostrum Intake

To estimate the level of colostrum intake in piglets, we used average daily gain at 24 h, as well as the concentration of IgG and α-tocopherol in plasma collected at 7 days of life (Table 2).

Split-suckling had a positive effect on weight gain during the first 24 h of life (*p* = 0.014), and there was an interaction between treatment and parity (*p* = 0.007), with split-suckling being more effective in the primiparous sows compared to the multiparous sows. There was no observed effect of sex on this parameter. No effect of treatment, parity, and sex on plasma α-tocopherol and IgG levels was observed, although this probability (*p* = 0.126) suggests a tendency of a higher α-tocopherol concentration in the piglets subjected to SS treatment (5.68 vs. 6.84 µg/mL).

The effect of BW and its interaction with SS treatment on daily gain at 24 h is shown in Figure 1. A marked effect of BW on weight gain is significant in all cases (*p* = 0.0002). Moreover, the slopes were different (*p* < 0.01) between experimental groups. There was a markedly affected weight gain for BW and SS.

Figure 2A shows the relationship between BW and plasma α-tocopherol concentration. Completely opposite trends were observed between experimental groups. Thus, CT showed that as BW increased, the plasma concentration of α-tocopherol increased, whereas the SS group showed a decreasing trend of α-tocopherol as the BW increased. The slopes were different (*p* < 0.05).

Figure 2B shows the relationship between BW and piglet plasma IgG concentration. In the CT group, a very slight upward trend in IgG content was observed with increasing live weight at birth. On the contrary, the trend observed in the split-suckling group was negative with increasing piglet birth weight. Also, the slopes were different in this case (*p* < 0.05).

### 3.3. Weight and Gains

Neither effect of T, P, S, nor the interactions were observed for birth or weaning weight (Table 3), but the piglets from the litters receiving the split-suckling treatment had a lower average daily gain during the lactation period (*p* < 0.001) than the piglets from the control group. There was a significant effect of parity and sex on live weight at slaughter and average daily gain during the fattening period. Thus, the male piglets from multiparous sows presented higher values in both productive parameters than the female piglets from primiparous sows.

We used regression analysis to study the effect of BW and neonatal split-suckling treatment on live weight at weaning and slaughter (Figure 3). Best fitting was achieved with linear regression, which showed a significant effect of BW in both cases (*p* < 0.0001). Comparisons between slopes showed a difference response to treatment in weight at weaning (Figure 3A; *p* < 0.05) but not at slaughter (Figure 3B). The effect of treatment and birth weight on daily gain during lactation is shown in Figure 4. Also, in this case, difference in slope response was observed (*p* < 0.005).

### 3.4. Carcass and Meat Composition and Quality Characteristics

The effect of T, P, and S on carcass characteristics and splitting yield can be seen in Table 4. The pigs subjected to SS showed lower subcutaneous fat thickness and higher lean content. A marked effect of sex on subcutaneous fat content and carcass leanness was also observed, with the castrated males being fatter than the females (*p* < 0.0001). No effect of parity was observed on carcass characteristics, but an interaction between T and P was observed in all cases, in which the lower fatness in the pigs subjected to SS was more marked in the pigs from primiparous sows.

No differences related to the effect of split-suckling treatment on weight of main commercial cuts were observed, but ham muscle weight was higher in the pigs from MP sows. Also in this case, an interaction between T and P was observed for ham weight and ham muscle weight, in which the response to SS treatment on increasing ham splitting yield was higher in the pigs from PM than in those coming from MP sows.

To study the effect of T in pigs according to BW in carcass and splitting yield data, we also carried out regression analysis. Comparisons between slopes showed a difference response to treatment only for ham subcutaneous fat (Figure 5), thus indicating a more marked effect of SS on the low-BW piglets.

The effect of T, P, and sex on the selected meat quality characteristics is shown in Table 5. A significant effect of parity was observed for color *L** value in LD and SM muscles. Moreover, SS treatment produced a lower color value in SM muscle (*p* = 0.044) with respect to the control group, but this effect was not observed in other muscles. No effect of T and P on IMF content was observed, but the castrated males showed higher IMF content than the females in LD (*p* = 0.028), and a similar trend in SM and BF muscles. Low BW showed a tendency for a higher concentration of intramuscular fat in LD muscle (*p* = 0.055) than the high-BW pigs.

A major effect of sex on main fatty acids of IMF was observed. Concentration of C18:2n-6 was higher in females and concentration of MUFA and SFA was higher in barrows. The effect of BS and SS treatment on fatty acids was limited. There was a higher concentration of C18:3n-3 (*p* = 0.036) and a tendency towards higher concentration of C18:2n-6 (*p* = 0.107) and unsaturation index (UI) (*p* = 0.113) and lower concentration of C16:0 (0.053) and SFA (*p* = 0.064) in those pigs that received the SS intervention. An interaction between T and BW was observed for C18:0 (*p* = 0.048) in which the effect of treatment in low BW pigs was of higher magnitude (12.8 for CT vs. 12.2 for SS) than in high-BW pigs (12.6 for CT vs. 12.7 for SS). This interaction was also observed for Δ9-desaturase activity index (*p* = 0.036), the effect of treatment being also more marked in low BW pigs (55.2 for CT vs. 56.0 for SS) than in high-BW pigs (55.4 for CT vs. 54.8 for SS). A tendency of TxBW for SFA (*p* = 0.063) was also observed.

## 4. Discussion

### 4.1. Mortality

At weaning, mortality was significantly affected by BW, with the low-BW piglets showing a four-fold higher mortality percentage than the piglets of normal weights, mainly concentrated within the first days of life, which is in agreement with the literature [18], and is attributed to differences in body reserve factors, including glycogen [19] and subcutaneous fat [20].

At weaning, SS treatment led to a moderate increase in mortality compared to the control group. This result may be due to excessive human intervention, which may negatively affect sow-piglet interaction during the initial critical neonatal period. No differential effect of SS treatment was observed according to BW, which suggests that increased mortality probably comes also from the heaviest piglets in the litter. A possible explanation may be that those piglets suffered more from being separated from the mother. This result contrasts with that of Donovan and Dritz [21], where no differences were observed between the study groups. In contrast, Holyoake et al. [22] reported a decrease in mortality of low-BW piglets at an average weaning age of 21 days.

### 4.2. Colostrum Intake

In this study, weight gain during the first 24 h of life of piglets subjected to split-suckling was higher than those of the CT group (Table 5). Morton et al. [7] did not find differences between groups. In addition, Vandaele et al. [8] reported the opposite effect of our study, i.e., split-suckling piglets showed a lower ADG value (24 h) than the control group. These different results may be due to the different split-suckling protocols used in each study. Moreover, differences in mortality (which affected mainly low-BW piglets) may also produce a bias, as dead piglets’ data do not account for weight gain calculation. Therefore, these results require further studies to reach an accurate conclusion at this stage of lactation. On the other hand, our study agrees with that of Morton et al. [7], in that piglets with higher BW have a higher ADG at the initial stage of lactation than smaller ones, thus reflecting higher colostrum intake.

Plasma concentration of α-tocopherol and IgG concentration is often utilized as a procedure to estimate piglet colostrum intake. In the control group, we observed a lack of effect of BW on piglet plasma IgG or α-tocopherol concentration which agrees with the literature [7,23]. This result contradicts differences in weight gain, which suggest higher colostrum intake in normal-BW piglets. A possible explanation is that bigger piglets require higher colostrum intake to achieve a certain plasma level of α-tocopherol and IgG. On the other hand, in our study, a negative effect of BW on plasma IgG concentration was observed in the split-suckling group; thus, the heavier piglets reached lower IgG concentrations. This result agrees with Alonso et al. [10] for sows of more than one farrowing. Another possible effect that may interfere with the results may be caused by the variability in colostrum IgG content between sows [24].

The concentration of α-tocopherol in piglet plasma may also be a suitable indicator as a measure of colostrum intake, as piglets are born with a low α-tocopherol content, and colostrum is a good source of this nutrient [25]. It is interesting to note that the trend of response is similar in IgG and α-tocopherol concentration (Figure 2).

The reason for the decrease in both IgG and α-tocopherol in plasma in the heavier piglets subjected to SS treatment may be due to the split-suckling protocol, which may be too intrusive for heavier piglets and may affect them negatively. These results suggest that an adequate split-suckling protocol should take into consideration possible negative effects on heavier (and most valuable) piglets, but more information is needed to establish plasma IgG levels for adequate piglet protection.

### 4.3. Weight and Gains

We observed that a difference in live weight of 100 g at birth produced differences of 300 g approximately at weaning (Figure 3A) and 11 g/d in daily weight gain during lactation (Figure 4). These results agree with the literature, where a significant effect of BW on weaning weight was observed [4,6,10,20,26]. Moreover, we also observed a positive correlation between BW and daily gain in later stages of growth, and consequently, on final weight. A difference of 100 g at birth produced differences of approximately 2 kg of live weight at slaughter (Figure 3B). This is also in agreement with the available literature [27,28,29].

The piglets from litters receiving the SS treatment showed a lower average daily gain during the lactation period (*p* < 0.001) than the piglets from the control group (Table 3). Some authors reported beneficial effects of split-suckling provided during the first day of suckling on neonatal growth [7] and mortality [13], whereas others observed detrimental long-term effect on growth if this treatment is applied for 3 d [8]. Differences may be due to SS protocols, which may be based either on birth weight or birth order. In each case, a variety of ways to carry out this management procedure have been used, with marked differences in the selection of animals, separation time and duration length of the intervention (from a few hours to several days). All these factors made it difficult to compare results and to extract clear conclusions. In comparison to others, our procedure may be considered “mild”, as there was only 1 h of separation time, and the total intervention time lasted for just 3 h. The study of the different response to SS according to BW revealed that in heavy pigs, SS may markedly reduce colostrum intake and weaning weight gain along with lactation (Figure 3A). These results agree with those reported by Vandaele et al. [8] who used 1.5 h of separation time but with an intervention length of 3 d. Moreover, Arnaud et al. [30], using a similar protocol to ours (six of the heaviest pigs removed from the sow for 1 h, reintroduction for 1.5 h, and removal again for 1 h), observed that SS decreased colostrum intake, ADG from birth to weaning, and body weight at weaning in high-birth-weight piglets (>1.25 kg). These results indicate that the potential benefit of SS on low-BW piglets is countered by a marked negative effect on high-BW piglets, thus producing an overall negative effect of the litter on weaning weight.

No differences in weaning weights were observed at the end of fattening (Table 3; Figure 3B). Although studies to test the effect of SS on live weight at slaughter are scarce, the available literature reported no effect on slaughter or carcass weight [30]. Split-suckling enhances colostrum intake in low-BW piglets but does not increase their growth in later phases of growth.

### 4.4. Carcass and Meat Composition and Quality Characteristics

As expected, the carcasses from the castrated males were fatter than those of the females (*p* < 0.0001) [31], but no interaction of sex was observed with SS treatment or parity.

The pigs that received SS treatment showed lower subcutaneous fat thickness (*p* < 0.005) and higher lean content (*p* < 0.005), this effect being more marked in the pigs from primiparous sows (*p* < 0.005 and 0.0005, respectively) (Table 4). There is limited information on the effect of neonatal management in piglets and its effect on carcass and meat quality. In recent research, Romero et al. [28] reported that neonatal care, which includes activities such as tying the umbilical cord, drying and massaging, eliminating mucus and debris from nasal and oral surfaces, and carefully putting the mouth into a functional nipple, increased lean content in carcasses from low-BW piglets. More recently, Arnaud et al. [30] reported that split-suckling reduced fat depth (13.1 mm vs. 14.4 mm; *p* < 0.01) and increased lean meat yield (58.8% vs. 58.0%; *p* = 0.02) in pig carcasses. Piglets with low BW, especially those classified as having intrauterine growth retardation (IUGR), are more susceptible to obesity [32], but the effect of split-suckling on increasing leanness of low-BW piglets is a novel finding, and of potential interest in neonatal swine. One potential explanation could be linked to the metabolic stimulation and neonatal imprinting effect of certain components in colostrum. These components may regulate biological processes including adiposity, insulin sensitivity, prevention of oxidative stress, lipogenic activity in white adipocytes, and sex differentiation among others [33].

IMF concentration was higher in low-BW piglets in LD muscle. Romero et al. [28] also observed a higher concentration of IMF in low-BW pigs. A limited effect of SS was observed in meat quality characteristics (pH, color, IMF) but the fatty acid composition was modified due to SS treatment (Table 5), which reinforces the idea of alteration in adiposity and metabolic regulation in the low-BW piglets receiving high levels of colostrum. In the pigs that received SS treatment, a higher concentration of C18:3n-3 (*p* = 0.036) and a tendency towards a higher concentration of C18:2n-6 (*p* = 0.107) and unsaturation index (UI) (*p* = 0.113) was observed in intramuscular fatty acids at slaughter, together with a lower concentration of C16:0 (*p* = 0.053) and SFA (*p* = 0.064). In finishing pigs, the concentration of unsaturated fatty acids is correlated to leanness, whereas the concentration of SFA (particularly C16:0 and C18:0) is associated with fatness [34]. Interestingly, an interaction between T and BW was observed for C18:0 (*p* = 0.048) in which the effect of treatment in the low-BW pigs was of higher magnitude than in the high-BW pigs, thus again indicating a differential response of SS treatment according to BW. This interaction was also observed for Δ9 desaturase activity (*p* = 0.036), the effect of treatment also being more marked in low-BW pigs. Remarkably, in pigs, a positive correlation has been reported between IMF and MUFA contents [35], both parameters considered as essential due to reported technological and nutritional needs [36,37].

This research was carried out under a productive commercial setting, and some questions that may be of importance, such as sire effect, were not fully controlled (heterospermic insemination was used).

## 5. Conclusions

Split-suckling increases opportunities for low-birth-weight piglets to obtain sufficient amounts of colostrum but may produce negative effects on heavier (and most valuable) piglets that gain less weight during lactation and receive a lesser amount of colostrum. Mortality is not affected by split-suckling. Split-suckling increases carcass leanness and alters fatty acid composition in pigs at slaughter. Long-term response to SS, particularly in low-birth-weight piglets, suggests alteration in adiposity and metabolic regulation in low-BW piglets receiving high levels of colostrum.

## Figures and Tables

**Figure 1 animals-13-03521-f001:**
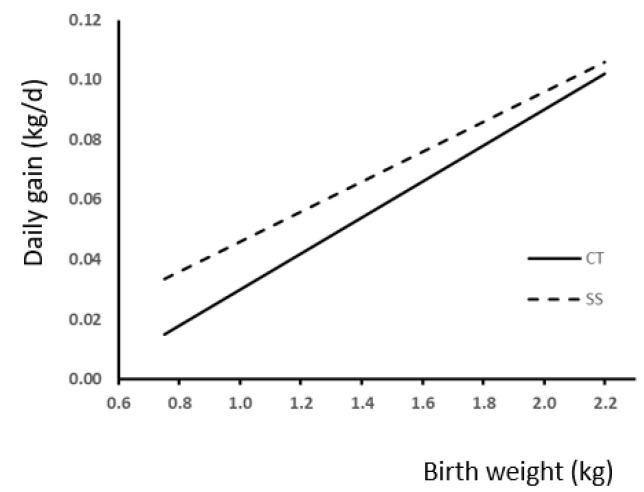
Relationship between birth weight and weight gain during first 24 h of life because of the presence (SS) or absence (CT) of split-suckling (weight gain 24 h (kg) in SS group = 0.05 (±0.014) × BW–0.004 (±0.018); R^2^ = 0.04; *p* = 0.0002; Weight gain 24 h (kg) in CT group = 0.06 (±0.014) × BW–0.03 (±0.018); R^2^ = 0.05; *p* = 0.0002).

**Figure 2 animals-13-03521-f002:**
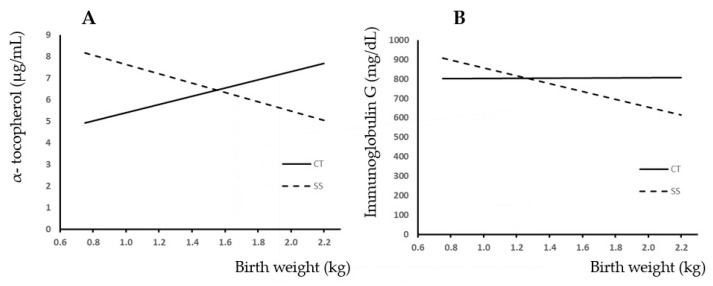
Relationship between birth weight and plasma α-tocopherol (**A**) and plasma IgG (**B**) because of the presence (SS) or absence (CT) of split-suckling. CT group: Vit E (μg/mL) = 1.9 (±1.211) × BW + 3.49 (±1.64); R^2^ = 0.035; *p* < 0.12; IgG (mg/dL) = 3.77 (±88.74) × BW + 799.53 (±117.38); R^2^ = 0.01; *p* < 0.966. SS group: Vit E (μg/mL) = −2.16 (±2.07) × BW + 9.79 (±2.81); R^2^ = 0.02; *p* < 0.3; IgG (mg/dL) = −202.98 (±119.39) × BW + 1060.13 (±185.78); R^2^ = 0.06; *p* < 0.096.

**Figure 3 animals-13-03521-f003:**
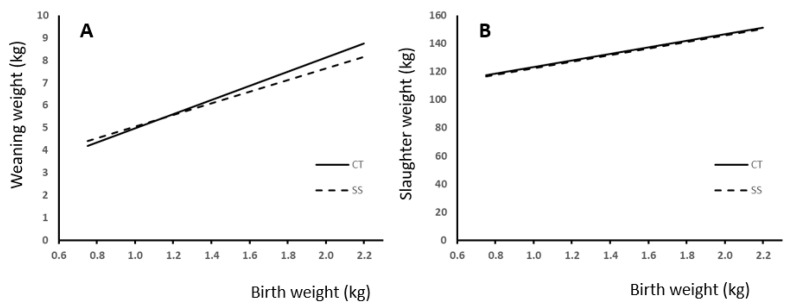
Relationship between birth weight and (**A**) weaning or (**B**) slaughter weight and the difference response according to the presence (SS) or absence (CT) of split-suckling treatment. (A—weaning weight: CT group (kg) = 3.12 (±0.287) × BW + 1.87 (±0.397); R^2^ = 0.33; *p* < 0.0001; SS group (kg) = 2.57 (±0.323) × BW + 2.50 (±0.45); R^2^ = 0.2; *p* < 0.0001). B—slaughter weight: CT group (kg) = 23.49 (±3.56) × BW + 99.68 (±4.92); R^2^ = 0.18; *p* < 0.0001; SS group (kg) = 23.26 (±4.05) × BW + 99.05 (±5.71); R^2^ = 0.15; *p* < 0.0001).

**Figure 4 animals-13-03521-f004:**
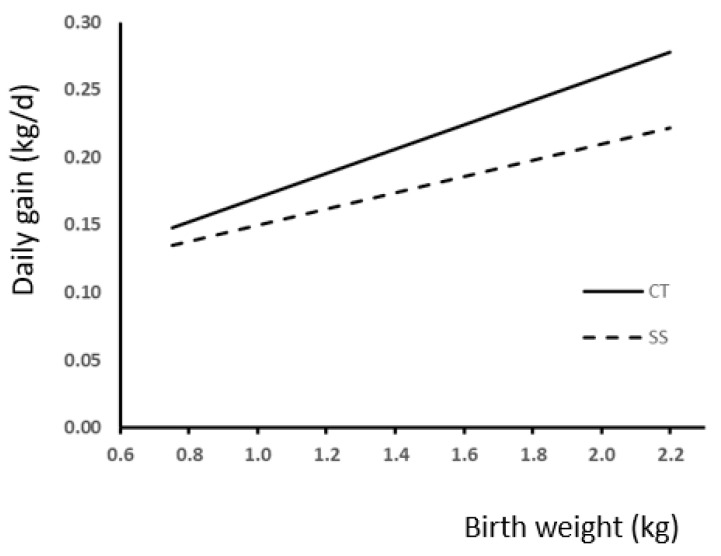
Relationship between birth weight and daily gain during the lactation period according to the presence (SS) or absence (CT) of split-suckling. (CT group, ADG (kg/d) = 0.09 (±0.013) × BW + 0.08 (±0.018); R^2^ = 0.17; *p* < 0.0001; SS group, ADG (kg/d) = 0.06 (±0.013) × BW + 0.09 (±0.018); R^2^ = 0.08; *p* < 0.0001).

**Figure 5 animals-13-03521-f005:**
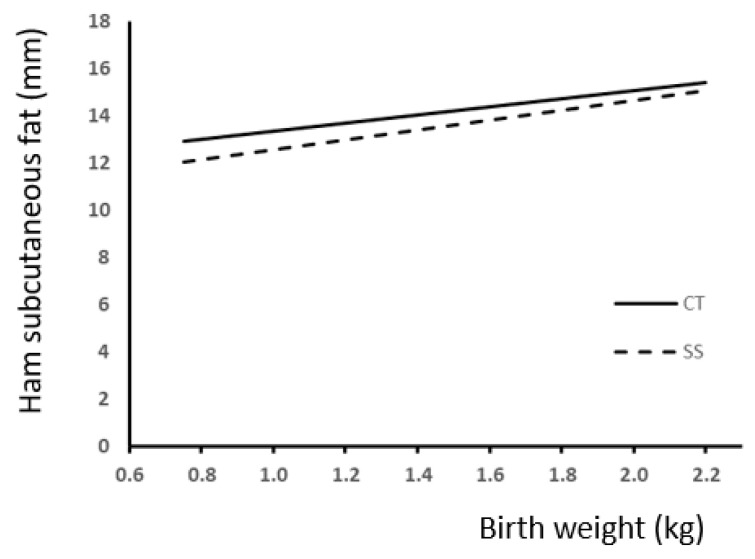
Relationship between birth weight and ham subcutaneous fat (SCF) accumulation according to the presence (SS) or absence (CT) of split-suckling. (CT group, SCF (mm) = 1.73 (±0.72) × BW + 11.6 (±0.99); R^2^ = 0.17; *p* < 0.01; SS group, SCF (mm) = 2.07 (±0.86) × BW + 10.5 (±1.21); R^2^ = 0.14; *p* < 0.01).

**Table 1 animals-13-03521-t001:** Effect of birth weight (BW), split-suckling treatment (T), parity (P), and sex (S) on weaning mortality rate analyzed by Chi-square comparison (low BW: weight equal to or less than 0.95 kg; normal BW: weight over 0.95 kg; CT: control; PP: primiparous; MP: multiparous; M: male; F: female).

	Birth Weight (BW)		Treatment (T)		Parity (P)		Sex (S)		*p*-Value			
	Low	Normal	CT	SS	PP	MP	M	F	BW	T	P	S
6d	0.304	0.047	0.08	0.086	0.067	0.090	0.075	0.092	0.0001	0.805	0.356	0.427
Weaning	0.554	0.157	0.171	0.25	0.247	0.201	0.212	0.215	0.0001	0.016	0.204	0.928

**Table 2 animals-13-03521-t002:** Effect of treatment (T), parity (P), and sex (S) on three indirect estimators of colostrum intake (piglet daily gain during the first 24 h of life, plasma α-tocopherol and immunoglobulin G concentration at 7 days of life) (CT: control; SS: split-suckling; PP: primiparous; MP: multiparous; CM: castrated male; F: female).

	T		P		S		SD	P < F	T	P	S	TxP	TxS
	CT	SS	PP	MP	CM	F							
Gain 0–24 h (kg)	0.05	0.07	0.07	0.05	0.05	0.06	0.074	0.000	0.015	0.011	0.154	0.007	0.405
Plasma α-toc (µg/mL)	5.68	6.84	5.82	6.70	6.77	6.18	3.391	0.405	0.126	0.264	0.906	0.105	0.251
Plasma IgG (mg/dL)	811	790	825	776	812	780	371.4	0.503	0.763	0.673	0.941	0.687	0.457

**Table 3 animals-13-03521-t003:** Effect of treatment (T), parity (P), and sex (S) on live weight (birth, weaning and slaughter) and average daily gain (lactation and fattening periods) (CT: control; SS: split-suckling; PP: primiparous; MP: multiparous; CM: castrated male; F: female).

	T		P		S		SD	P < F	T	P	S	TxP	TxS
	CT	SS	PP	MP	CM	F							
Weight (kg)													
Birth	1.27	1.31	1.26	1.31	1.34	1.27	0.1103	0.0001	0.4841	0.4913	0.4361	0.4215	0.1435
Weaning	6.06	6.01	5.99	6.08	6.11	5.99	1.4191	0.0001	0.1817	0.6591	0.3587	0.501	0.9011
Slaughter	129	129	125	133	137	126	14.379	0.0001	0.4076	0.0119	0.0005	0.175	0.6027
Daily gain (kg/d)													
Lactation	0.20	0.18	0.20	0.19	0.19	0.19	0.0603	0.0001	0.0012	0.0866	0.2356	0.1072	0.8515
Fattening	0.74	0.74	0.71	0.76	0.78	0.72	0.0842	0.0001	0.5548	0.0059	0.0004	0.1654	0.5622

**Table 4 animals-13-03521-t004:** Effect of split-suckling treatment (T), parity (P), and sex (S) on carcass characteristics and splitting yield (CT: control; SS: split-suckling; PP: primiparous; MP: multiparous; CM: castrated male; F: female; LS means, Cov: covariable carcass weight; SC: subcutaneous).

	T		P		S		SD	P < F	T	P	S	TxP	TxS	Cov
	CT	SS	PP	MP	CM	F								
Carcass														
SC fat (P2, mm)	19.2	17.9	18.5	18.5	19.7	17.33	3.12	0.0001	0.0028	0.9580	0.0001	0.0006	0.8558	0.0001
SC fat (Ham, mm)	14.2	13.3	13.7	13.8	14.625	12.85	2.19	0.0001	0.0013	0.6143	0.0001	0.0034	0.7065	0.0001
% Lean	57.8	58.8	58.3	58.3	57.15	59.43	2.73	0.0001	0.0027	0.8051	0.0001	0.0002	0.9764	0.0001
Splitting yield (kg/kg carcass)														
Loin	3.18	3.17	3.16	3.19	3.115	3.238	0.22	0.0001	0.8986	0.3809	0.0011	0.5923	0.8051	0.0001
Ham	13.3	13.4	13.3	13.4	13.25	13.4	0.403	0.0001	0.4831	0.1644	0.0801	0.0009	0.6811	0.0001
Ham muscles	7.38	7.41	7.26	7.53	7.2225	7.565	0.409	0.0001	0.7653	0.0044	0.0004	0.0042	0.2905	0.0001

**Table 5 animals-13-03521-t005:** Effect of birth weight (BW), treatment (T), parity (P), and sex (S) on selected meat quality characteristics (L: <1.4 kg; H: >1.4 kg; CT: control; SS: split-suckling; PP: primiparous; MP: multiparous; CM: castrated male; F: female; UI: unsaturation index).

	BW		T		P		S		SD	P < F	BW	T	P	S	TxBW	TxP	TxS
	L	H	CT	SS	PP	MP	CM	F									
pH and sacoplasmic protein solubility																	
pH25 (loin)	6.7	6.7	6.7	6.7	6.7	6.7	6.6	6.7	0.19	0.252	0.755	0.410	0.458	0.612	0.946	0.525	0.215
pHu (loin)	5.4	5.3	5.4	5.4	5.4	5.4	5.39	5.35	0.09	0.019	0.010	0.816	0.345	0.007	0.562	0.742	0.668
Sarc. Pr.Sol.	12.6	14.8	14.2	14.4	13.2	15.5	14.3	14.7	8.38	0.003	0.126	0.084	0.114	0.082	0.094	0.439	0.002
Intramuscular fat																	
LD	2.9	2.6	2.9	2.6	2.8	2.8	3.0	2.5	0.84	0.078	0.055	0.622	0.712	0.028	0.200	0.646	0.591
BF	2.9	2.9	3.1	2.8	3.1	2.9	3.2	2.7	0.58	0.037	0.233	0.435	0.339	0.074	0.378	0.715	0.091
SM	3.9	3.6	4.1	3.7	4.0	3.7	4.2	3.6	0.92	0.045	0.139	0.494	0.217	0.264	0.521	0.737	0.056
*a* value																	
LD	1.9	2.3	2.1	2.0	2.0	2.1	2.1	2.0	0.84	0.057	0.090	0.409	0.420	0.466	0.699	0.380	0.452
BF	1.9	13.9	14.1	13.8	13.8	14.0	13.9	13.9	1.23	0.524	0.677	0.169	0.800	0.726	0.552	0.281	0.566
SM	13.5	13.7	13.7	13.2	13.6	13.4	13.4	13.5	1.26	0.791	0.838	0.044	0.504	0.993	0.258	0.258	0.894
*b* value																	
LD	13.1	13.3	13.2	13.1	13.2	13.1	13.1	13.1	1.01	0.014	0.058	0.134	0.141	0.950	0.059	0.177	0.886
BF	13.4	13.3	13.6	13.1	13.4	13.3	13.4	13.3	1.12	0.788	0.617	0.137	0.108	0.837	0.752	0.778	0.655
SM	14.1	14.0	14.4	13.6	14.1	14.0	14.0	14.1	0.99	0.231	0.662	0.086	0.118	0.788	0.902	0.196	0.557
*L* value																	
LD	70.4	70.0	70.5	70.1	70.7	69.9	70.3	70.3	2.27	0.288	0.118	0.143	0.010	0.622	0.689	0.491	0.773
BF	51.8	51.4	52.0	51.1	51.9	51.2	51.5	51.5	2.65	0.930	0.459	0.191	0.073	0.842	0.785	0.777	0.951
SM	54.7	54.4	54.8	54.2	54.7	54.2	54.5	54.5	2.28	0.655	0.542	0.641	0.045	0.589	0.932	0.491	0.680
Intramuscular fatty acids (loin)																	
C16:0	24.0	23.7	24.1	23.6	23.9	23.8	24.4	23.4	1.02	0.001	0.655	0.053	0.488	0.000	0.350	0.894	0.758
C18:0	12.5	12.6	12.7	12.4	12.6	12.6	12.8	12.3	0.92	0.017	0.192	0.219	0.189	0.010	0.048	0.145	0.216
C18:1 n-9	42.4	41.6	42.3	41.9	42.4	41.8	42.1	42.0	2.03	0.361	0.153	0.777	0.220	0.938	0.371	0.872	0.920
C18:2 n-6	8.0	8.8	7.9	8.8	8.2	8.5	7.7	8.9	1.83	0.041	0.290	0.107	0.325	0.008	0.923	0.578	0.778
C18:3 n-3	0.4	0.4	0.36	0.38	0.4	0.4	0.35	0.39	0.06	0.010	0.135	0.036	0.287	0.010	0.936	0.199	0.685
SFA	38.1	38.0	38.5	37.7	38.1	38.0	38.9	37.3	1.71	0.000	0.697	0.064	0.802	0.010	0.063	0.395	0.672
MUFA	50.4	49.4	50.2	49.7	50.2	49.7	50.1	49.8	2.28	0.322	0.128	0.623	0.144	0.990	0.470	0.964	0.913
PUFA	11.4	12.6	11.3	12.6	11.7	12.2	11.0	12.9	2.92	0.044	0.303	0.141	0.319	0.010	0.889	0.641	0.738
Δ9-Desat	55.6	55.1	55.3	55.5	55.5	55.3	54.9	55.8	1.6	0.004	0.281	0.166	0.339	0.000	0.036	0.628	0.626
UI	79.2	81.5	78.8	81.8	79.7	80.9	77.8	82.6	6.36	0.012	0.427	0.113	0.440	0.010	0.635	0.604	0.665

## Data Availability

Data are contained within the article.

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
