# Peer review of "Short- and Long-Term Effects of Split-Suckling in Pigs According to Birth Weight"

_animals, 2023, doi:10.3390/ani13223521_

Round 1

Reviewer 1 Report

Comments and Suggestions for Authors

An interesting paper that is well crafted with a strong methodology, results and discussion. It is interesting that the authors suggest that split-suckling is become common in the industry, despite the lack of scientific evidence; especially given the substantial labour input involved. I wonder what methodologies are commonly used to apply this technique in industry – the authors mention that the literature includes a variety of timings, but I cannot see a justification for the choice of timings used in this specific study.

In terms of the results, I found that the tables had a lot of information in them, in particular the detailed p values. I am a little old fashioned and prefer to see the significant values called out, with the non-significant values simply notated with “ns” or similar – to my mind that makes it easier to see the important values in these tables. However I am happy to leave that to the editors to determine their preferred display method.

I also find the use of “tendency” in terms of some of the results to be a little tenuous. P values between 0.1-0.05 might be described in that way to the reader, but values >0.1 are non-significant and could be treated as such?

One question does come to mind as I read the results – how long was each sow farrowing for? How much variation in farrowing length was there. Given that the split-suckling was a relatively short period of time, variation in farrowing length could allow piglets born early in the process a significant extra amount of time on the udder, prior to intervention.  How was the “end” of farrowing established? Again later born piglets may have not been as active as earlier born piglets at that point in time.

Specific comments

There were no line numbers on the manuscript copy, I have tried to accurately locate my comments for the authors.

Abstract – a double full stop just before “Weight gain during the first 24 hours of life …”

Tendencies – see earlier general comment

Introduction – There appears to be a font change in the “it is not easy to estimate” paragraph.

You have a sentence starting with “On the other hand” – it is not clear to me how this relates to the previous sentences, I would simply delete that phrase

Methods – “did not imply any invasive”. I am not sure how the word imply is correct here. I might suggest “impose”?

2.2 – “along gestation”. “During gestation…” may be a better word?

The methods talk about the removal of “normal BW piglets” but we do not see any definition of “normal” prior to Table 1.

Results – Table 2. Units for gain?

Figure 2A and 2B – given that it is the slope of the line that is of interest, do the two points at each end add much to the figure, or do they distract? Or imply only two points were used to plot the line. Again this may be a convention the editors wish to comment on. That same comment obviously applies to the other figures drawn in the same way.

In addition, figure 2 does not include the extra detail in its title such as R value (compare to figure 1 for example).

Table 3 and 4 – trying to understand the notation P<F in the table (significance?)

Discussion – pre-weaning mortality. It is hard to judge exactly the effects here without knowing the causes of the latter mortality during lactation. Most early mortality is relatively consistent as the authors point out, but late stage mortality could be more variable. It may also be that split-suckling keeps the weaker piglets alive for longer, compared to traditional suckling were they die earlier. i.e. postponing the inevitable.

The discussion on IgG relates to my question on farrowing length – not only are sows variable, but we know that variation exists within the udder and levels fall rapidly during the first day of suckling. A long vs short farrowing could alert uptake – one would hope that the number of litters used here would average out any effects prior to the treatment being imposed.

References

Number 21 is incomplete

Number 29 and 32 have Journal of animal science with lower case letters whilst others have Journal of Animal Science with upper case letters.  

Author Response

Thank you for your comments. You can see the answers to your requirements in the attached document.

Reviewer 2 Report

Comments and Suggestions for Authors

Please read the comments in the reviewed document.

Chi-square analysis needs justification.

Comments on the Quality of English Language

Overall language quality is good, some sentences require rephrasing (highlighted in the reviewed document). In discussion, few results are written in present form, need to be revised in past sentences.

Author Response

(The authors gave the same response as above.)

Reviewer 3 Report

Comments and Suggestions for Authors

Review report for the manuscript “Short- and Long-Term Effects of Split-Suckling in Pigs according to birth weight”

Brief summary

The study focused on effects of split-suckling and birth weight on performance on pigs on a commercial farm. The results indicated the necessity of further studies to confirm these findings

General concept

The methodology needs to be elaborated particularly aspects of the study design- ethics approval, data collection and analysis. Tabulated results are difficult to understand.

Specific comments

Title

Lines 2-3: It is not clear what the title “Short- and Long-Term Effects of Split-Suckling in Pigs according to birth weight” means. Please consider a shorter version e.g. “Short- and Long-Term Effects of Split-Suckling in Pigs

Line 14: Should "reduced" be "reduces"?

Line 15: Consider replacing "produce" with "lead to"

Line 16: consider changing "small piglets" to either "piglets with a low birth weight" or "low-birth-weight piglets"?

Line 17: Please mention what the period is. For example "short period during suckling"

Line 19 -20: change "sucking" to "suckling". Please either maintain or drop the hyphen between "split" and "suckling"

Line 21: change "along" to "during". Consider "cohort" or "contemporaries" or "liter mates" in place of "corresponding control"

Line 22: refer to comment at line 19

Line 24: refer to comment at line 19

Line 26: It would be interested to know if any study has reported on the efficiency of colostrum utilization by low-birth-weight piglets. This is important considering that SS has no effect on piglet mortality rate. In other words should it be recommended?

Line 28: Some detail on litter size would be informative. Otherwise the average for the litters may be assumed but this is sensitive to outliers.

Line 30-31,33: Please reconsider the choice of abbreviations over the full word especially for the abstract

Line 35: should "concentrated" be "highest"? Does this results in part explain why SS had no effect on piglet mortality rate since other factors were at play and could have denied SS sufficient time to take effect?

Line 36-37: Given that weaning date was variable, could this have confounded the effects of SS? This result needs to be elaborated.

Line 37-40: In what way was SS more effective- weight gain, piglet survival?Could mothering ability be confounding SS?

Line 40-41: was the ADG computed at piglet or litter level? I think it is important to clarify that irrespective of birth weight, piglets in the control litters had higher ADGs or it was the liiter ADG that was higher.

Line 41-43: It is important to note that high and low wright piglets were in both treatment and control groups and therefore, I suggest that the authors emphasize that differences were irrespective of piglet weight, for, example as reported in the next sentence.

Line 44-46: Was this irrespective of piglet body weight? Was there significant interaction between treatment x body weight?

Line 46-48: ""SS was not a significant effect on slaughter weight"

Line 47-49: It is important to justify analysis of pooled data rather than by class of piglet weight.

Line 50: what is LD- " longissimus dorsi muscle"?

Line 54: No conclusion?

Line 58: change "advances" to "improvements"

Line 60: Is this a standard weight or a cutoff e.g 1 kg? Is it determined from the average litter body weight? Change "related to" to "influenced by"

Line 61-62: make "which is

considered one of the causes of higher preweaning mortality in hyperprolific sows [4]." a sentence e.g. "Inadequate colostrum intake is

considered one of the causes of higher preweaning mortality in hyperprolific sows [4]."

Line 66-67: Please break into shorter concise sentences.

Line 69: Is "later stages of the piglet" meant to be "later in life" or "in adult life"?

Line 73: Should "peripartum" be "postpartum"?

Line 74-77: Please rewrite for clarity

Line 77: Please delete "The determination of" and provide a reference

Line 82: Please change "part of the litter" to "some litter mates". Also change "less favored" to "low-body weight"

Line 103: Please confirm that the competent authority approved this study. Given the requirement for compliance with the EU directives, please state the authority that confirmed this. This should be in addition to the approval granted by the Animal Welfare Committee.

Line 112: Please explain the preference for heterospermic insemination. Given that the authors advance that their results inform the practical applications of SS, is the use of heterospermic insemination regular farm practice? Will it have an effect on the results of the study?

Line 113: Please provide more details of the sires- how many? Purebred?

Line 116: Please change "Along" to "During". Please explain the groups of 60 considering that only 48 sows were used in the study

Line 118: Please add "the" between "to" and "expected"

Line 132-135: Please elaborate this. Were the piglets given to sows other than their dam? Why was this done? What was the distribution of birth weight that the authors sought to maintain. These need to be clearly explained.

Line 138: Please quantify this

Line 139-140: How does this relate to line 132-135? As i asked earlier, was the decision to remove piglets based on their body weight? The authors suggested this in Line 92-93. This needs to be clarified.

Line 138-147: Please clearly explain the rationale of steps 1-4. The SS strategy needs to be clearly presented. What was the litter size and how was the membership of the litters determined? Given that the piglets were tagged, how then were they classified as heavy or low birth weight? Why was the separation period from the sow one hour? Did this separation stress the piglets and/ or sow? Which piglets in SS were nor separated from the sow for the duration of the study?

Line 148: Why was it necessary to weigh the piglets on day 1?

Line 149-150 should be linked to line 132-135.

Line 150-151; Please provide rationale given the classification based on birth weight

Line 159: should "independently of previously assigned treatments" be "in both the treatment and control groups"

Line 163: Please explain what this "All replicate pens had a similar average BW" means. If possible quantify this e.g. "the average BW of piglets in all replicate pens was mean+SD

Line 167: What are the qualifications of the experienced staff? Veterinary assistants, stockmen,, nutritionists, technicians?

Line 169: Is this true for piglets and sows?

Line 174: How many pigs? 645?

Line 192-196: Please explain how this fits into the experimental design described earlier in this section.

Line 227: Please re-write this section and clearly present the approach followed. Two statistics are presented- Chi-square test and GLM. Please elaborate each separately. Clearly describe the independent and dependent variables and the choice of models used.

Line 229: Please confirm that "chi-square procedure" is a SAS procedure e.g. " PROC GLM"

Line 231: Please provide the correct SAS nomenclature for "general linear model (GLM) procedure contained in SAS" and all other procedures.

Line 232: Should "productive" be "production"?

Line 234: Please refer to other sections for symbols used here. Split-suckling as SS not T. Should "birth split-" be "birth weight, split-"?

Line 235: Please explain what "splitting yield covariance" is.

Line 252-253: How did this SS effect validated? Several factors affect pre-weaning survival in addition to the BW. How were the SS effects teased out?

Line 256: What is meant by "mortality ration"?

Line 259: This table is not clear to me. Please unpack this table.

Line 266: Is "weight gain" supposed to be "Average Daily Gain"?

Line 267: Please refer to my comment on description of variables. Was parity a class variable? If the interaction is significant, why then is the conclusion focused on SS?

Line 270-271: Why make this conclusion while the level of significance is predetermined?

Line 273: This table is not clear to me. What do the numbers mean in terms of the title of the table (effects)?

Line 284: Please maintain abbreviations, symbols, names throughout the manuscript

Line 309: were the males multiparous?

Line 311: All female pigs that have farrowed are called sows

Line 313: Please format the tables to communicate a clear message. I see a disconnect between the table and its title.

Line 344: castration has a positive effect on body weight. How were these effects separated from those of sex?

Line 346-347: How was mothering ability accounted for?

Line 464: Change "along" to "during"

Line 565: Given the plausible confounder in the pre-weaning period, it is difficult to draw firm conclusions from the study. This is especially true for performance later in life and indeed additional research is required..

Line 586: The IRB is one of the initial approvals granted and the final approval should be granted by the nationally mandated agency. Please clarify this.

Line 589: The study was conducted on a farm whose owner should have provided prior consent

Comments on the Quality of English Language

/

Author Response

Thank you for your suggestions. We believe that the manuscript has been considerably improved by them.

kind regards.

Round 2

Reviewer 2 Report

Comments and Suggestions for Authors

The authors have addressed all the concerns, it may be published as such.

Author Response

Thank you for your comments. We are very grateful that the article has improved considerably thanks to the recommendations made by you.

Reviewer 3 Report

Comments and Suggestions for Authors

The clarity of the manuscript has greatly improved. Apart from the typos, the quality of the manuscript is suitable for publication in animals.

Comments on the Quality of English Language

The clarity of the manuscript has greatly improved. Apart from the typos, the quality of the manuscript is suitable for publication in animals.

Author Response

(The authors gave the same response as above.)
